# Intelligent Driver Assistance and Energy Management Systems of Hybrid Electric Autonomous Vehicles

**Ziad Al-Saadi [1], Duong Phan Van [2], Ali Moradi Amani [1], Mojgan Fayyazi [1], Samaneh Sadat Sajjadi [1], Dinh Ba Pham [2], Reza Jazar [1] and Hamid Khayyam [1,*]**

[1] School of Engineering, RMIT University, Melbourne, VIC 3083, Australia;
s3638468@student.rmit.edu.au (Z.A.-S.); ali.moradiamani@rmit.edu.au (A.M.A.);
mojgan.fayyazi@rmit.edu.au (M.F.); samaneh.sadat.sajjadi@rmit.edu.au (S.S.S.); reza.jazar@rmit.edu.au (R.J.)

[2] Division of Mechatronics, Mechanical Engineering Institute, Vietnam Maritime University,
Haiphong 180000, Vietnam; duongpv.vck@vimaru.edu.vn (D.P.V.); bapd.vck@vimaru.edu.vn (D.B.P.)

[*] Correspondence: hamid.khayyam@rmit.edu.au or hamid.khayyam@ymail.com

**Abstract:** Automotive companies continue to develop integrated safety, sustainability, and reliability features that can help mitigate some of the most common driving risks associated with autonomous vehicles (AVs). Hybrid electric vehicles (HEVs) offer practical solutions to use control strategies to cut down fuel usage and emissions. AVs and HEVs are combined to take the advantages of each kind to solve the problem of wasting energy. This paper presents an intelligent driver assistance system, including adaptive cruise control (ACC) and an energy management system (EMS), for HEVs. Our proposed ACC determines the desired acceleration and safe distance with the lead car through a switched model predictive control (MPC) and a neuro-fuzzy (NF) system. The performance criteria of the switched MPC toggles between speed and distance control appropriately and its stability is mathematically proven. The EMS intelligently control the energy consumption based on ACC commands. The results show that the driving risk is extremely reduced by using ACC-MPC and ACC-NF, and the vehicle energy consumption by driver assistance system based on ACC-NF is improved by 2.6%.

**Keywords:** intelligent energy management; adaptive cruise control; autonomous vehicle; model predictive control; artificial intelligence; complex systems

## 1. Introduction

Autonomous vehicles (AV), i.e., vehicles that are derived by computers, are coming to our roads [1,2]. It is anticipated that companies will have a USD 7 trillion annual revenue stream from the AVs market in 2050 [3]. They are supposed to bring us more safety, relaxation, and sustainability than traditional vehicles. Congestions, fuel consumption, and $CO_2$ emissions are also expected to be reduced by AVs. Adaptive cruise control (ACC) is one of the main parts of AVs, which controls the vehicle acceleration based on the driving style. Sensors, such as radar, lidar, or a camera are used to observe the road and inform ACC about the relative distance and speed to the leading vehicle. ACC keeps a desired distance from the leading vehicle by adjusting the throttle and/or the brake system automatically. In a reasonable driving condition, a vehicle equipped with ACC travels at a driver-set velocity by controlling the throttle, similar to the operation of conventional cruise control. In the case of detecting a lead vehicle, the ACC system performs calculations to determine whether the vehicle is still able to travel at a set-speed. If the measured distance between the two vehicles is less than the safe distance, the ACC sends appropriate signals to the engine or braking system to decelerate the vehicle.

The global commitment toward sustainability [4] is expected to stimulate investments in new technologies. The transportation sector is notoriously known as one of the major contributors in increasing $CO_2$ emissions worldwide. The feasibility study of an electrified

transportation system via hybrid electric vehicle (HEV) and electric vehicle for zero or low carbon emission has been studied [5]. These emission figures are directly proportional to the energy consumed by the same sector. According to the Australian Department of the Environment and Energy (2019) report, the road transport sector was responsible for 19% of overall energy consumption in Australia. Mostly from passenger and light commercial vehicles, which are solely driven by ICE.

The desire to reduce carbon emissions due to transportation sources has led over the past decade to the development of new propulsion technologies, including hybrid, plug-in hybrid, and battery electric vehicles [6]. With more stringent laws and policies to reduce $CO_2$ emissions and mandates to decrease the dependence on fossil fuel resources, automotive manufacturers are developing new technologies and design concepts to meet the new laws and regulations. Electric vehicles have been introduced, but they still face many challenges, such as limited travel range and lack of existing charging stations infrastructure to accommodate the increase in their numbers. Thus, different implementations of hybrid vehicles have been developed as an alternative to the full induction of electric vehicles to the roads. The plug-in HEV is one of the most successful implementations of HEV. It has the suitable configuration to charge the battery when there is an available charging station, as well as adding fuel to ICE when electric charge is not charging station, as well as adding fuel to ICE when electric charge is unavailable. Moreover, the battery can also be charged during engine braking or directly from the ICE [4].

AVs can be powered in the form of hybrid electric vehicles (HEV)s, which are driven using a combination of an electric motor and the conventional internal combustion engine (ICE) [7]. An HEV has many benefits over a pure electric vehicle (EV) in terms of travelling range and convenience, as the battery onboard can be charged automatically without the need for a charging station. It can consume considerably less fuel compared to the traditional ICEs [7]. Despite these advantages, energy management problems emerge in HEVs to ensure the efficient operation of the battery [4].

HEVs are divided into three categories: series (S-HEV), parallel (P-HEV), and series/parallel (S/P-HEV). Synchronizing multiple power sources and controlling optimal power flow between mechanical and electrical parts has less complexity in P-HEVs than other types [8]. Therefore, P-HEV is considered in this paper, for which the power flow is illustrated in Figure 1 as a combination of ICE, powertrain, motor, and pack of batteries.

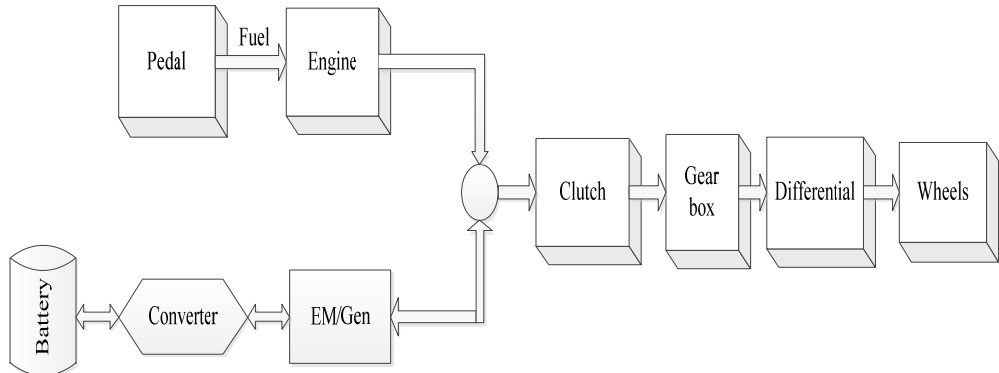

**Figure 1.** Composition of a P-HEV.

The automotive powertrain is a complex, a highly non-linear, and a time-varying dynamical system by nature [2]. Several algorithms have been proposed to solve the ACC problem, such as a PID controller [9], a look-ahead control system [10], a linear quadratic control optimal synthesis approach [11], a recurrent cerebellar model articulation controller [12], and a model predictive control [13–15]. Augmenting the EM to the vehicle's powertrain makes it even more sophisticated to control. Intelligent and learning-based algorithms show capabilities to deal with these complexities [16–21]. Although not straightforward, fuzzy control can translate the experiences and knowledge of expert drivers to

if-then rules. A tuning process is required to obtain the optimal parameters and membership functions for the fuzzy system. This process can be managed using a neuro-fuzzy inference system. By embedding a neural network into the traditional fuzzy control system, the membership functions and if-then rules can be optimized using the training dataset. The biggest challenges for autonomous vehicles are: (i) make decisions faster in very diverse conditions, (ii) important potential in reducing pollution by optimizing their routes, (iii) considerable gap between the self-drive technology level and the current regulations, (iv) safety and imminent accidents, and (v) cyber security to defend against attacks [22,23].

This paper proposes an intelligent driver assistance system that includes a neuro-fuzzy (NF)-based EMS combined with a switched MPC system for ACC. The controller is designed to maintain a safe distance between two successive cars while the energy consumption for the following vehicle, which is a P-HEV, is reduced. The ACC controller switches between a speed and distance control mode based on the vehicle condition. This problem falls in the switched MPC category, which has been widely studied in the control community [24,25]. Asymptotic stability of the propose switched MPC is proven mathematically. The controller's performance is investigated under combined load scenarios, including driver behaviour, environment conditions, and vehicle specifications. The proposed controller is also compared with a model predictive controller (MPC)-based ACC in safety and energy saving. It is shown that the proposed controller can better manage nonlinearities in the energy saving of a P-HEV, resulting in less fuel consumption.

The rest of the paper is formatted as follows: Section 2 presents the problem formulation, Section 3 introduces the vehicle configuration and road power demand of the ego car, Section 4 presents the proposed driver assistance system for the ego car, followed by Section 5 illustrating the simulation results and discussion. Section 6 shows the conclusions.

## 2. Problem Formulation

Figure 2 illustrates the scenario considered in this paper where an ego HEV follows a lead vehicle in the same lane. $x_l$, $v_l$, and $a_l$ symbolize the position, velocity, and acceleration of the lead car, respectively. Those of the ego car are represented as $x_e$, $v_e$, and $a_e$, respectively. The ego P-HEV is equipped with an ACC and appropriate sensors to measure the relative distance $d = x_l - x_e$ and relative velocity $v = v_l - v_e$. Figure 4 depicts the designed driver assistance system for the ego car, including an ACC and an EMS. The spacing error $\Delta d$, and the relative speed $v$ are defined as,

$$\Delta d = d - d_{safe} \tag{1}$$

$$v = v_l - v_e \tag{2}$$

The control objectives are considered as follows:

- The ACC is aimed to maintain a safe car following distance while the ego car follows the speed of the lead car. That means $\Delta d = \epsilon > 0$ for a small $\epsilon$ and $v \to 0$;
- The EMS should reduce the energy consumption of the ego car.

Generally, an ACC controller is proposed to be hierarchical, including an upper-level controller and a lower-level controller [26]. The upper-level controller typically regulates the desired acceleration for the vehicle based on the relative speed and relative distance measured to the lead car in the same lane (as shown in the ACC block in Figure 4). The lower-level controller regulates the throttle and brake to follow the acceleration/deceleration from the upper-level controller. In this paper, the upper-level controller is emphasized, so the lower-level controller is assumed to be well-designed.

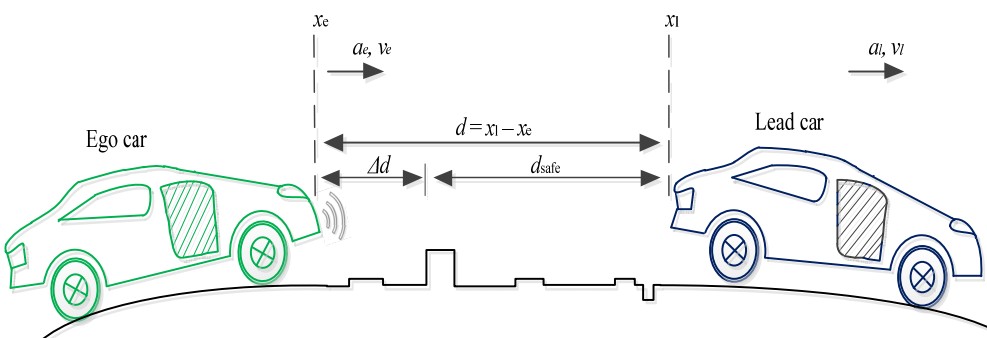

**Figure 2.** Illustration of considered scenario.

## 3. Vehicle Configuration and Road Power Demand

Before starting the controller design process, the modelling of different parts of the P-HEV is reviewed here.

### 3.1. Internal Combustion Engine

Figure 3 depicts a typical internal combustion engine. Considering the air mass entering the manifold (mi), the manifold pressure (Pm), and the engine speed (ωe) as the states and the throttle angle (α) as the input signal, the following nonlinear state-space equations describe the dynamics of ICE [27]:

$$\dot{x}_1(t) = MAX \cdot \left[1 - e^{9\left(\frac{x_2(t)}{P_{atm}} - 1\right)}\right] \left[1 - \cos(1.14459u(t) - 1.06)\right]$$
$$\dot{x}_2(t) = -\frac{V_e\eta_{vol}}{4\pi V_m}x_2(t)x_3(t) + \frac{MAX.R.T}{V_m} \cdot \left[1 - e^{9\left(\frac{x_2(t)}{P_{atm}} - 1\right)}\right]\left[1 - \cos(1.14459u(t) - 1.06)\right] \quad (3)$$
$$\dot{x}_3(t) = \frac{C_T V_e \eta_{vol}}{4\pi I_e RT} \cdot AFI(t - \lambda) \cdot SPI(t - \delta) \cdot x_2$$

in which $x(t) = [x_1, x_2, x_3] = [mi, Pm, \omega e]$ is the state vector, $u(t) = \alpha(t)$ is the engine input signal, and $0 \leq \alpha \leq 79.46°$. The following parameters are considered for the engine [27]: $MAX = 0.1843$ Kg/s, $V_e = 0.0038$ m3, $V_m = 0.0027$ m$^3$, $AFI = SPI = 1$ and $\lambda = \delta = 0$, i.e., no delay is considered. Despite these simplifications, Equation (3) is still a sophisticated nonlinear system to control. The mass fuel rate consumption can be represented as static functions of engine torque $\tau_e$ and the speed of engine $\omega_e$.

$$\dot{m}_{fuel} = \frac{\omega_e \cdot \tau_e}{q_c \cdot \eta_m \cdot \eta_e} \quad (4)$$

where $q_c$ is the combustion energy, $\eta_m = 0.9$ [28] is the mechanical efficiency, and $\eta_e$ is the engine efficiency.

### 3.2. Electric Motor

An electric motor generates extra torque on the crankshaft when required. It can also work in the generation mode to generate electricity from the crankshaft rotation and charge the battery. The torque generated by the EM can be calculated as,

$$\tau_m = \alpha_e \tau_{e-max} \quad (5)$$

where $\alpha_e$ and $\tau_{e-max}$ are throttle electronic signal and the max motor torque, respectively. The supplied battery power $P_b$ is calculated as follows [29].

$$P_b = \frac{\tau_m \cdot \omega_m}{9550 \ \eta_m} \quad (6)$$

where $\eta_m$ is the efficiency of the motor, $\tau_m$ is the output torque of the motor, $\omega_m$ is the motor speed, and $P_b$ (kW) is the power of the battery pack.

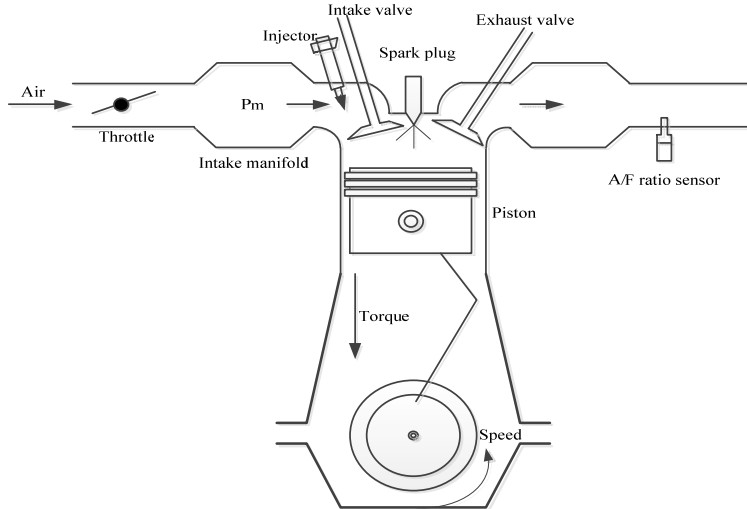

**Figure 3.** A typical internal combustion engine.

### 3.3. Battery

The battery in P-HEV acts as both an energy storage and a source of power. The battery charge capacity $Q$ illustrates the maximum power stored that can be stored in the battery. A battery's state-of-charge ($SoC$) is defined as the ratio of the remaining charge over its full capacity; thus, $0 \leq SoC \leq 1$ where $SoC = 1$ means the battery is fully charged. Battery discharge results in the $SoC$ varying with the rate,

$$\frac{d(SoC)}{dt} = -\frac{U_0 - \sqrt{U_0^2 - 4P_bR}}{2QR} \tag{7}$$

where $U_0$ is open circuit voltage, $R$ is battery internal resistance, $Q$ is battery charge capacity. $P_b$ can be negative (battery absorbs power from the ICE) and positive (battery provides power to drive the powertrain). To obtain the best performance and prolong the lifecycle of a battery, the $SoC$ is typically limited to an interval, which can be described as follows.

$$SoC_{min} \leq SoC(t) \leq SoC_{max} \tag{8}$$

### 3.4. Road Power Demand

The total required power of a vehicle is determined as [8],

$$\begin{aligned}
P_{RPD} &= P_{rf} + P_{dg} + P_{slope} + P_{acc} + P_d = C_{rolling}mg\cos\theta \cdot v_t \\
&+ C_{drag(\varphi)} \cdot \tfrac{1}{2}\varphi(v_w + v_t)^2 A(\varphi) \cdot v_t + mg\sin\theta \cdot v_t + MC_{room}\frac{dT_{room}}{dt} + mv_t\frac{dv_t}{dt}.
\end{aligned} \tag{9}$$

where $m$ is the vehicle mass, $\theta$ depicts road slope, $M$ presents the air mass inside the cabin, Troom is the air temperature in the cabin, $A(\varphi)$ is front surface area, $v_w$ is the absolute wind velocity, $v_t$ is the speed of the vehicle, $C_{rolling}$ is the road friction coefficient, $C_{drag}$ is the drag coefficient, and $\rho$ is the air density. All the parameters exploited to estimate the road power demand are shown in Table 1.

**Table 1.** Vehicle specifications and parameters are used to calculate the vehicle's required power.

| Specification | Parameters | Value |
|---|---|---|
| Road friction coefficient | $C_{rolling}$ | 0.015 |
| Gravity acceleration | $g$ | 9.81 m/s$^2$ |
| Vehicle velocity | $v_t$ | ACC command m/s |
| Wind velocity | $v_w$ | m/s |
| Mass (vehicle + equivalent rotating parts + passengers) | $m$ | 1280 kg |
| Drag coefficient (constant) | $C_{drag}$ | 0.335 |
| Front surface area | $A$ | $1.9 \times (1/\cos\phi)$ |
| Air density | $\rho$ | 1.225 kg/m$^3$ |
| Combustion energy | $q_{combustion}$ | 38017 kJ/kg |
| Wheel radius | $wh_r$ | 0.285 m |
| Differential ratio | $d_r$ | 3.21:1 |
| Electric motor/generator | | |
| Maximum current | | 480 A |
| Minimum voltage | | 120 V |
| Max power | | 75 kW |
| Battery pack | | |
| Chemistry | | Li-Ion |
| A cell nominal voltage | | 12 V |
| Nominal capacity | | 26.2 Ah |
| Pack battery power | $P_{Battery}$ | 4.4 kWh |
| Temperature | | [0 22 40] (0 °C) |
| Min voltage | | 9.5 V |
| Max voltage | | 16.5 V |

Driver behavior (the vehicle's speed) is obtained from the ACC system. Road and wind profiles are environmental conditions considered in this paper. The road data with characteristics mimicking the real roads is created by using the method presented in [30]. The Poisson distribution is utilized to develop the amount of road segments. The lengths of each road segment are obtained by using the exponential distribution. The Rayleigh distribution is employed to model the road's height up and down hills. The left and the right bends of the road are supposed to have a Gaussian distribution. A wind profile is also obtained from the model in [30]. It is a collection of sections of different lengths, wind speeds, and directions. The range, wind speed, and direction are modelled by using the exponential, Weibull, and uniform distribution models, respectively. The road power demand should be supplied by both ICE and EM [31]. It means,

$$P_{RPD} = P_{ICE}(t) + P_b(t) \tag{10}$$

Therefore, the fuel consumption rate of Equation (4) is updated to [29],

$$\dot{m}_{eqv} = \dot{m}_{fuel} + s(t) \cdot \frac{P_b}{H_{LHV}} \tag{11}$$

where $P_m$ is the battery power (kW) and $HLHV$ is the fuel lower heating value (kJ/kg). $s(t)$ is an equivalent factor defined as [29],

$$s(t) = -\frac{\lambda(t)H_{LHV}}{QU_0} \tag{12}$$

where $\lambda(t)$ is the co-state value that can be considered as an equivalent weight factor between fuel consumption and electrical power consumption. The optimal value of this factor is obtained from an iterative algorithm to fascinate the boundary of the *SoC*, and more details can be found in [32].

**4. Proposed Intelligent Power Driver Assistance System for the Ego Car**

Figure 4 depicts the designed driver assistance system for the ego car, including two parts: the ACC and the EMS. The control system is designed to ensure that the ego car maintains a safe distance from the lead car and increases its fuel economy.

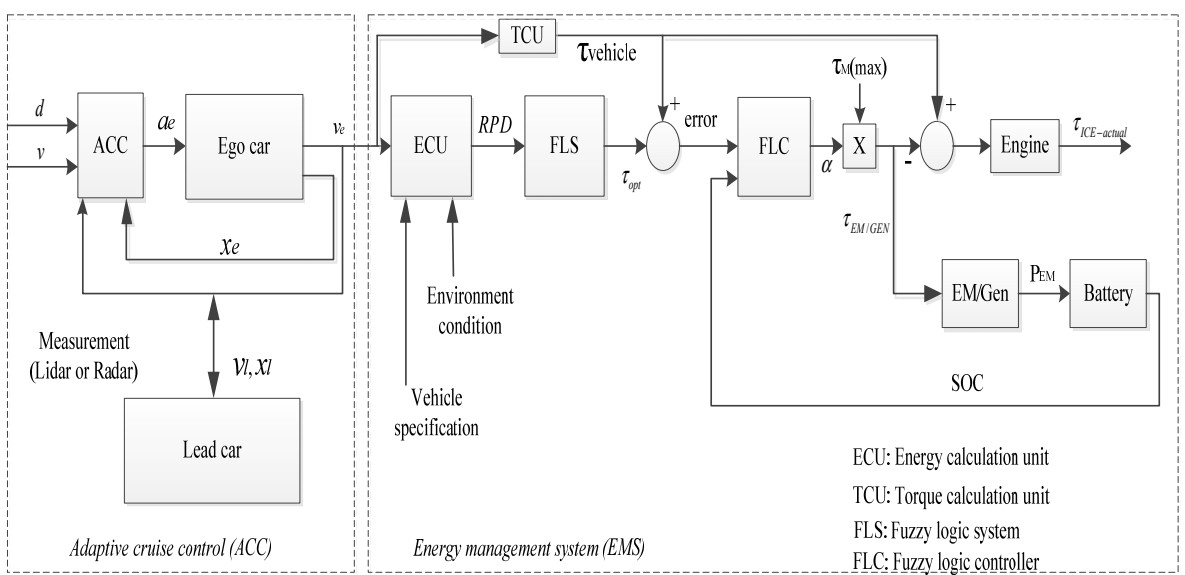

**Figure 4.** An intelligent driver assistant architecture.

*4.1. Adaptive Cruise Control*

4.1.1. Adaptive Cruise Control Based on Switched MPC

The ACC vehicle model can be defined as follows [13].

$$\tau\frac{da_e(t)}{dt} + a_e(t) = u(t) \tag{13}$$

where $\tau$ refers to the time lag depending on the finite bandwidth of the lower-level controller and $u$ depicts the acceleration command calculated from the upper-level control. By defining the state vector $z(t) = [\Delta d(t), \Delta v(t), a_e(t)]^T$, the state-space model of the equation of motion becomes

$$\dot{z}(t) = \begin{bmatrix} 0 & 1 & 0 \\ 0 & 0 & 1 \\ 0 & 0 & -\frac{1}{\tau} \end{bmatrix} z(t) + \begin{bmatrix} 0 \\ 0 \\ \frac{1}{\tau} \end{bmatrix} u(t) \tag{14}$$

MPC is a feedback control algorithm that uses the model plant to predict the future outputs of a process. It also uses the optimizer, which guarantees that the predicted plant output tracks the desired reference. By solving an optimization problem, the MPC controller tries to minimize the error between the reference and the predicted output over a future horizon, possibly subject to constraints on the manipulated inputs and outputs [33].

The control objective for the ego car is to maintain its speed close to the lead car while the relative distance is safe, i.e., $\Delta d \to 0$ and $v(k) \to 0$ as k reaches to infinity. A switched model predictive control (SAMPC) is used to achieve this objective. Acceleration of the ego vehicle should be adaptively changed in order to regulate $\Delta d$. The acceleration command is calculated by solving the following constrained optimization problem during each sampling period [33].

$$\underset{u}{\text{Min}}J(t) = \int_t^{t+T}\left\{z^T(t+k)Qz(t+k) + \Delta u^T(t+k)R\Delta u(t+k)\right\}dk$$

$$s.t\begin{cases} \Delta d \geq 0 \\ v_{min} \leq v_e(k) \leq v_{max} \\ a_{min} \leq a_e(k) \leq a_{max} \\ u_{min} \leq u(k) \leq u_{max} \end{cases} \tag{15}$$

where $t$ is the current time, $p$ is the prediction horizon, and $\Delta u$ is the increment of the control input. $Q_t$, $R_t^{\Delta u}$, and $R_t^u$ refer to the weight matrices for the following error, change rate, and magnitude of the control input, respectively.

As a normal ACC, the MPC control objective should be distance control, i.e., $z = z_1 = \Delta d$ in Equation (15). However, we will show in simulations that the performance of such a controller is poor when the ego car falls behind for any reason. In order to solve this issue, an AMPC is considered in which the control objective is adaptively changed based on the distance $\Delta d$ between ego and lead cars. When $\Delta d$ is large, i.e., the ego vehicle is far behind the lead one, AMPC switches to a speed control system. Therefore, $z = z_2 = \Delta v$ in the optimization problem (15), which results in accelerating the ego car to fill its gap with the lead. However, when $\Delta d$ becomes reasonably close to $d_{default}$, the control system switches to distance control and $z = z_1 = \Delta d$. In this case, the ego car follows the driving profile of the lead by increasing and decreasing longitudinal acceleration such that $\Delta d \to 0$.

This adaptive behavior makes the control algorithm robust against undesired disturbances. For example, if the ego car fails in tracking $\Delta d \to 0$ for any reason, such as a sudden action of the lead driver or the loss of sensor signals, it will be easily compensated by switching to the speed control mode for a while.

The performance of the ACC based on MPC will be discussed in Section 5.

### 4.1.2. Stability of the Proposed Controller

Several research activities in switching MPC (SMPC) have been addressed by allowing switching between controllers, e.g., [34]. However, in many cases, such as the one we have in this paper, switching between different performance criteria in different operating conditions may be required. In both cases, the stability of the switching system should be studied.

Suppose that an MPC controller should be designed for the dynamical system

$$\dot{x} = f(x, u); x(0) = x_0 \tag{16}$$

with a Lipschitz function $f$ such that the performance criteria

$$J_p(x, u) = \int_{t_k}^{t_k+T} L_p(x(\tau), u(\tau))d\tau + F_p(x(t_{k+T})) \tag{17}$$

is optimized in which $L_p$ is a continuous positive definite function, $F_p$ is a continuously differentiable positive definite terminal function, and $p \in \{1, 2, \ldots, P\}$ shows which cost function is now active. The following theorem is helpful in this study [34].

**Definition 1**. *The average dwell time $\tau a$ is the average number of time units between two consecutive switches.*

**Theorem 1** ([35]). *Assume that there exist constants $\mu \geq 1$ and $\lambda \geq 0$ such that for all p's we have*

$$V_{pi}(x) \leq \mu V_{pj}(x) \tag{18}$$

*and*

$$V_{pi}(x(t_2)) - V_{pi}(x(t_1)) \leq -\lambda \int_{t_1}^{t_2} V_{pi}(x(t))dt \tag{19}$$

Then, the closed loop MPC control system with the switching performance criteria is asymptotically stable if

$$\tau_a > \frac{ln\mu}{\lambda} \tag{20}$$

**Lemma 1**. *The state space equations control system (14) controlled by the switching MPC controller (15) is asymptotically stable.*

**Proof**. The performance criteria of Equation (15) switch between J1 and J2, corresponding to $z = z_1 = \Delta d$ and $z = z_2 = \Delta v$, respectively. We have,

$$V_1(z) = z_1^T(t+k)Qz_1(t+k) + \Delta u^T(t+k)R\Delta u(t+k)$$

$$V_2(z) = z_2^T(t+k)Qz_2(t+k) + \Delta u^T(t+k)R\Delta u(t+k)$$

The inequality (18) is satisfied for $\mu = 1$ because

$$z_1^T(t+k)Qz_1(t+k) \leq z_2^T(t+k)Qz_2(t+k)$$

$$\Delta d^T(t+k)Q\Delta d(t+k) \leq \Delta v^T(t+k)Q\Delta v(t+k)$$

$$0 \leq \left[\Delta v^T(t+k) - \Delta d^T(t+k)\right]Q[\Delta v(t+k) - \Delta d(t+k)]$$

which is satisfied due to the positive definite matrix $Q$. This shows that condition (18) is satisfied in our problem for $\mu = 1$, which indeed results in $\tau_a > 0$ from Equation (20). From Theorem 1, one can conclude that the asymptotic stability of the system is satisfied for any dwell time [35]. □

4.1.3. Adaptive Cruise Control Based on NF

The NF system exploited in this paper is the adaptive-network-based fuzzy inference system (ANFIS) proposed in [16]. Due to the inherent complexity, nonlinearity, and uncertainty under multi-scale responses, energy management control of hybrid electric autonomous vehicles is required to use intelligent control such as a fuzzy logic system or a neural network. As ANFIS is the combination of a neural network and fuzzy logic, and it gives accuracy to non-linear systems' adaptation capability and rapid learning capacity, it is widely being used [36]. ANFIS control has been used in a wide range of engineering applications such as electrical engineering, mechanical engineering, chemical, robotics engineering, and many other engineering disciplines. It combines the artificial neural network with fuzzy logic control. This combination allows overcoming the limitation of employing fuzzy logic control as a stand-alone control technique. The fuzzy inference System (FIS) has the limitation that it is not an expert by itself. The reliability of FIS is highly dependent on the rules generated by the designer. These rules vary significantly based on the designers' expertise in the system. If an expert assigns the wrong rules, the FIS would probably perform poorly.

Moreover, in many FIS implementations, trial and error are used to improve the rules by the designer. Hence, in order to overcome these constraints using FIS alone, an artificial neural network can be used to make the FIS expert by itself. The artificial neural network

ANN uses the input–output data to learn about the system behavior, apply the correct rules, and assign the correct membership function values to obtain the best performance. ANFIS can learn and tune parameters in a fuzzy inference system by utilizing a hybrid learning algorithm. Fuzzy rules are extracted at each layer of a neural network. The Tankagi–Sugeno model is applied for the composition of if-then rules. ANFIS is composed of the five following layers, as depicted in Figure 5.

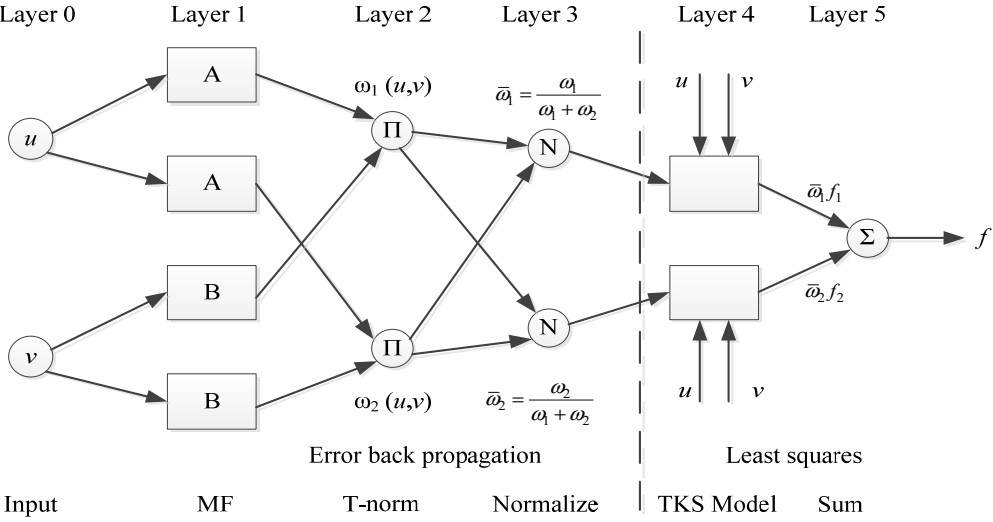

**Figure 5.** NF system with Tagaki–Sugeno inference.

Layer 1 [37]: The action of each node j in this layer is adaptable with a node function as follows.

$$O_{1,j} = \begin{cases} \mu_{A_j}(u), & j = 1,2 \\ \mu_{B_j}(v), & j = 3,4 \end{cases} \tag{21}$$

where $u$ (or $v$) is the input to node $j$, and $A_j$ (or $B_j$) is the linguistic variable related to the node membership function. $\mu_{A_j}(u)$, the membership function of $A_{j,}$ is represented as follows.

$$\mu_{A_j}(u) = \frac{1}{1 + [((u - c_j)/a_j^2)]^{b_j}} \tag{22}$$

where $u$ is the input and $(a_j, b_j, c_j)$ is the premise set.

Layer 2 [37]: Nodes in this layer perform a fixed function to calculate the product of all incoming signals.

$$O_{2,j} = \omega_j = \mu_{A_j}(u).\mu_{B_j}(v), j = 1,2 \tag{23}$$

The output signal $\omega_j$ illustrates the firing strength of a rule.

Layer 3: Nodes in this layer are also static (not adaptive). They are used to determine the proportion of the $j$-th rule's firing strength to the accumulation of firing strengths of all the rules. It means,

$$O_{3,j} = \bar{\omega}_j = \frac{\omega_1}{\omega_1 + \omega_2}, j = 1,2 \tag{24}$$

Layer 4: Nodes in this layer perform the following adaptive function.

$$O_4 = \bar{\omega}_j f_j = \bar{\omega}_j (p_j x + q_j y + r_j), j = 1,2 \tag{25}$$

where $\bar{\omega}_j$ is the output of layer 3 and $(p_j, q_j, r_j)$ is the sequential parameter set.

Layer 5 integrates all incoming signals to produce the overall output.

$$O_{5,j} = \sum \bar{\omega} f_j = \frac{\sum_j \omega_j f_j}{\sum_j \omega_j} \tag{26}$$

To work as an ACC in a P-HEV, we design an NF system that has two input signals: the relative distance and the relative velocity between two cars. It has an output, the acceleration signal applied to the ego car. Three membership functions are constructed for all input/output signals. Several experiments are performed to obtain the training dataset, which includes these signals. The training dataset is utilized for training the model by matching the expected output with the inputs, i.e., performing supervised learning. To anticipate the feedback for the model examinations after training, a validation stage is applied, followed by testing the method on a test dataset to produce an unbiased evaluation of the performance of the eventual model. At present, there is no specific solution to divide the data for these assignments [38]. However, based on [39], the set of the data is randomly divided into training (70%), validation (20%), and testing (10%). The performance results of the training, validation, and testing process are depicted in Figure 6. The FIS is generated using the grid partition method and the hybrid learning algorithm described in [37]. The model structure and the control surface are demonstrated in Figures 7 and 8. The membership functions of inputs that are automatically produced by the NF system are shown in Figures 9 and 10. The performance of the ACC based on MPC will be discussed in Section 5.

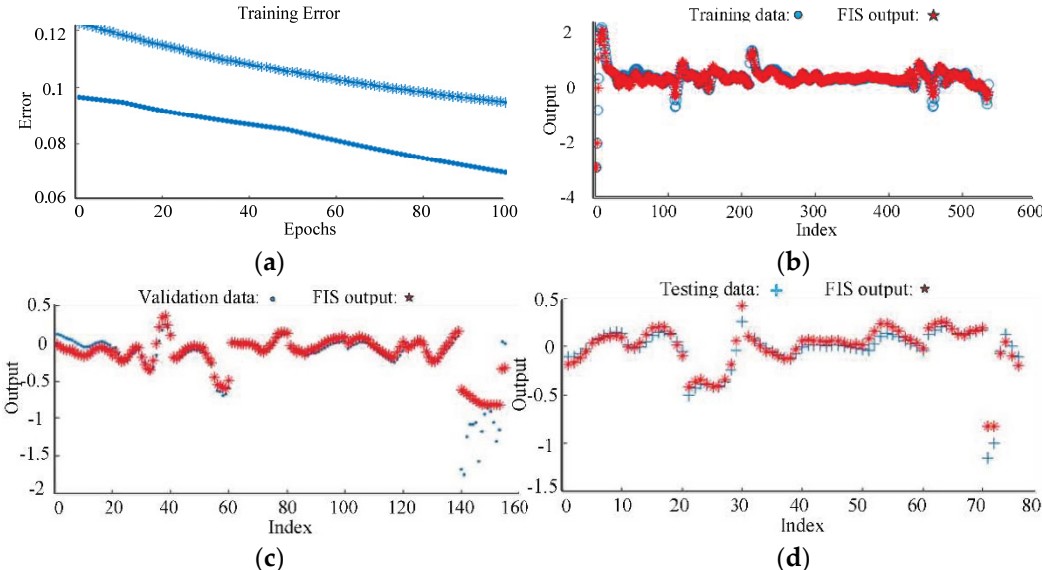

**Figure 6.** (**a**) Training error for 100 epochs with 0.094709 error tolerance; (**b**) Training data, average error was 0.9455; (**c**) Validation data, average error was 0.18137; (**d**) Testing data; average error was 0.06927.

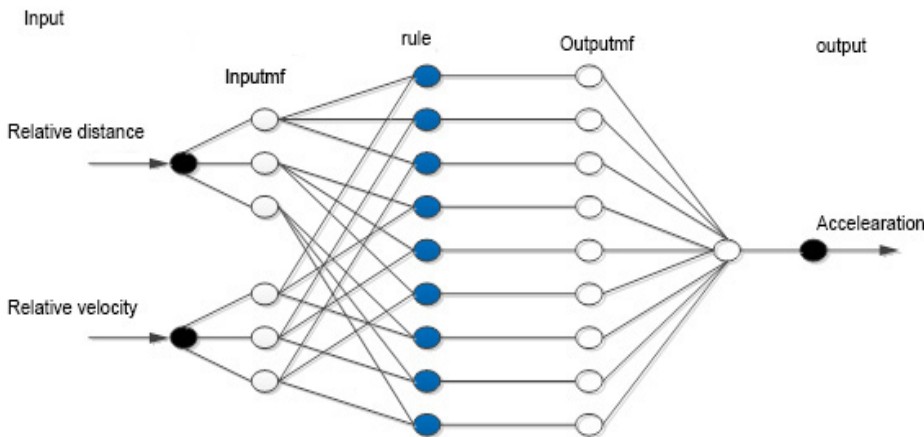

**Figure 7.** Model structure.

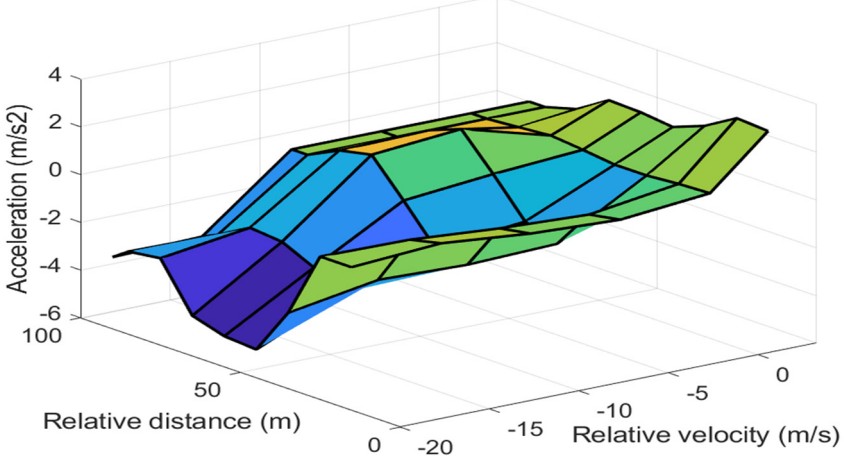

**Figure 8.** Control surface.

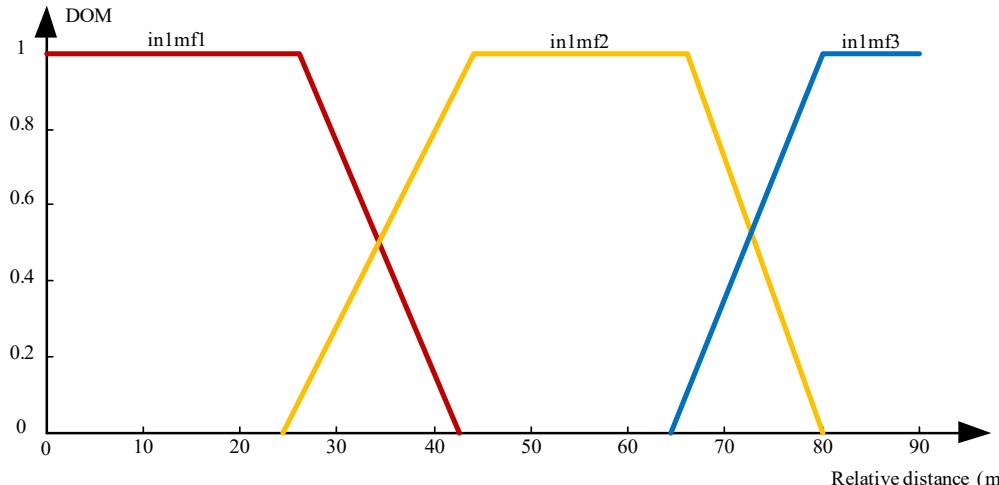

**Figure 9.** Membership function of input 1.

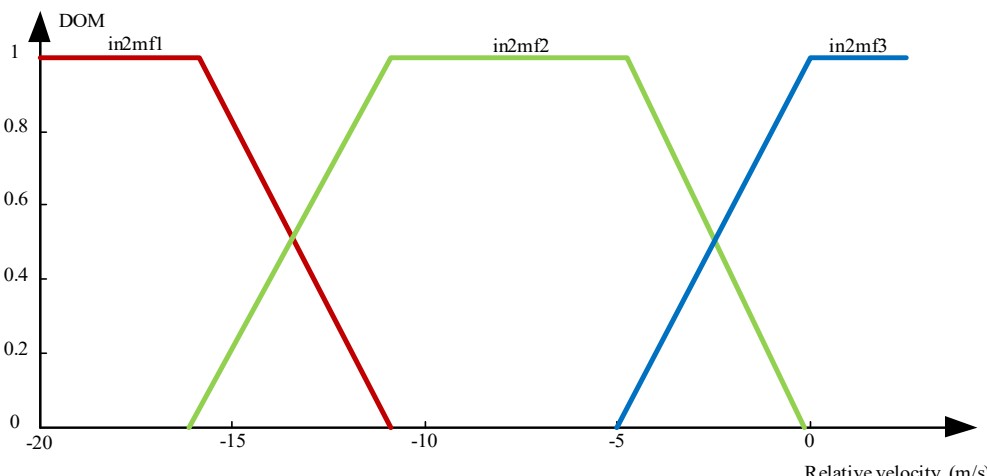

**Figure 10.** Membership function of input 2.

*4.2. Energy Management System*

The EMS is proposed to determine the power-split between the ICE and the electric motor based on the total required power of the vehicle. In the vehicle, the total essential power depicts a vehicle's fuel usage and determines how energy is exploited as the vehicle moves on the road. As shown in Figure 4, all the information about DEV (driver behaviour, environment conditions, and vehicle specification) is passed into ECU (energy calculation unit) to obtain the necessary power. It can be calculated based on Equation (9). Then, a TCU block (torque calculation unit) converts the information from the driver to the requirement for the torque, calculated as the following equation [8,40].

$$\tau_{vehicle} = \frac{\omega h_r \cdot P_{RPD}}{d_r \cdot g_r(t) \cdot v(t)} \tag{27}$$

Simultaneously, the fuzzy logic system (FLS) block produces the optimal torque to the ICE based on the total required power of the vehicle, as shown in [40]. The operation on this FLS block is as follows. The total necessary power is classified in five groups: VL, L, N, H, and VH. Fuzzy rules are proposed so the vehicle's engine can yield the torque close to the optimal torque area to escalate the engine performance. The output of the FLS defuzzifies the signal in three clusters: L, O, and H. By conducting many experiments, the fuzzification and defuzzification sections have been designed as Figure 11a,b, respectively.

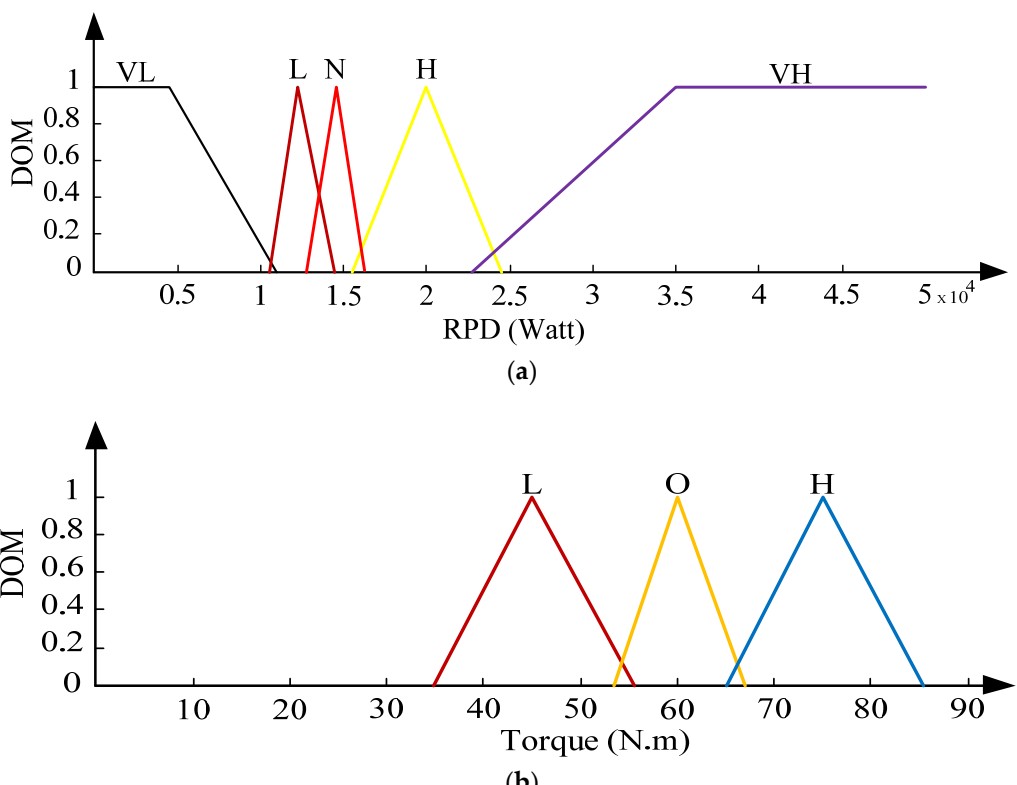

**Figure 11.** Membership functions of the fuzzy logic system: (**a**) input (**b**) output.

Using the generated membership functions above, the rule-based system has been generated experimentally. Table 2 shows the FLS rules that yield the optimal engine torque to ensure that the engine state is altered to perform at the maximum achievable efficiency.

Next, a fuzzy logic controller (FLC) is used to regulate the motor throttle electronic signal. Therefore, ICE will work at its optimal operating point. The rest of the required torque, which is called auxiliary torque in this paper, will be contributed by the motor. The throttle electronic signal is the output of the FLC block, which has two inputs (the error torque and the battery *SoC*), i.e., the fuzzy logic controller receives the error torque and the battery *SoC* and governs signal $\alpha$ to modify the motor torque.

**Table 2.** The rules of fuzzy logic system.

| Condition Number | If Total Required Power | Then $\tau_{optimal}$ |
|:---:|:---:|:---:|
| 1 | VL | L |
| 2 | L | L |
| 3 | N | O |
| 4 | H | H |
| 5 | VH | H |

The torque of the motor after the FLC block can be achieved by using Equation (5). Afterward, the torque of the engine can be estimated as follows.

$$\tau_{ICE} = \tau_{vehicle} - \tau_m \tag{28}$$

The signal $\tau_m$ belongs to one of the following groups: RT, T, TU, SU, C, or RC. We also consider three groups TBT, TB, and TBC for the state of charge (SOoC) of the battery. Fuzzy rules are designed so that the appropriate throttle electronic signal is generated to provide a suitable τm. Thus, the engine is able to be operated in the optimal area but still guarantees

not affecting the battery features. The output (throttle electronic signal) is divided into six clusters: RT, T, TO, SU, C, and RC. Figures 12 and 13 show the membership functions of inputs of the FLC, respectively.

Figure 14 represents the membership function of the FLC's output (throttle electronic signal). Table 3 describes the rules of the fuzzy logic controller.

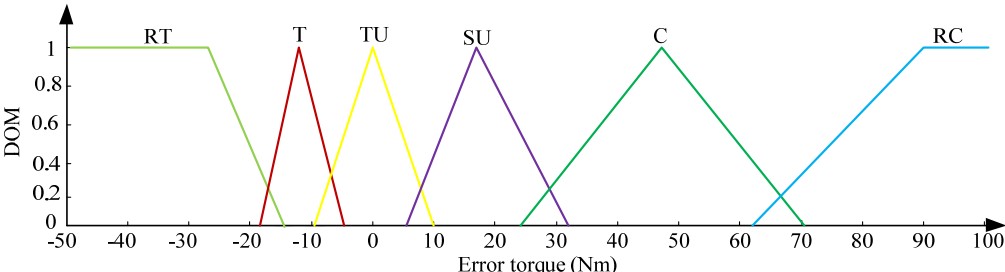

**Figure 12.** The difference between the requirement torque of the vehicle and the optimal torque of ICE.

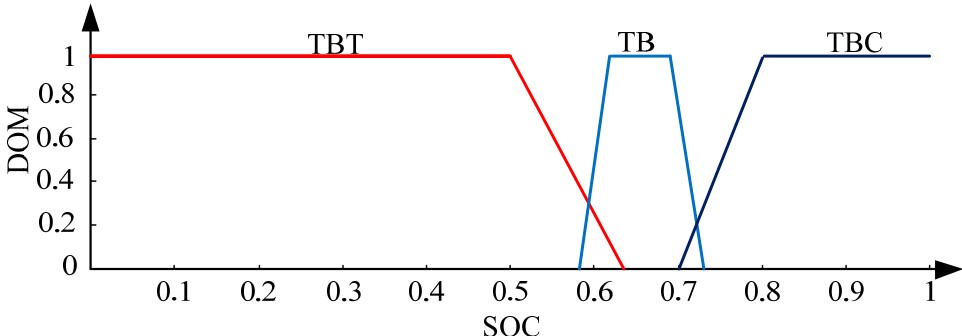

**Figure 13.** The state of charge of the battery.

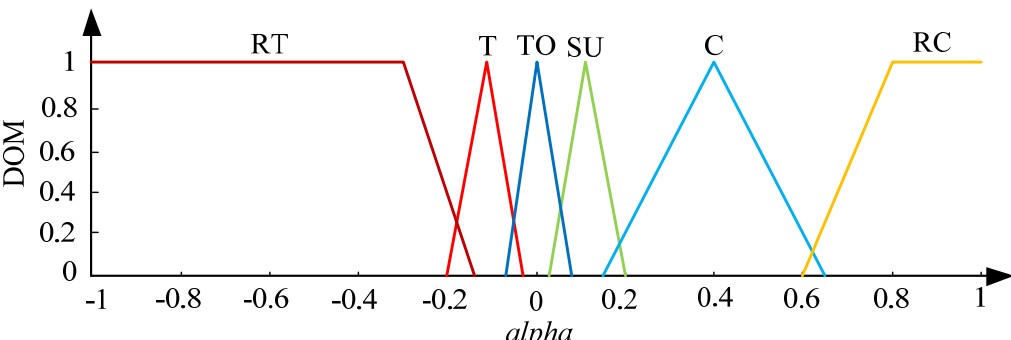

**Figure 14.** Throttle electronic signal.

**Table 3.** The rules of fuzzy logic controller.

| Condition Number | If $e_\tau$ | And $SoC$ | Then $\alpha$ | Condition Number | If $e_\tau$ | And $SoC$ | Then $\alpha$ |
|---|---|---|---|---|---|---|---|
| 1 | RC | TBT | TO | 10 | TU | TBT | TO |
| 2 | RC | TB | RC | 11 | TU | TB | TO |
| 3 | RC | TBC | RC | 12 | TU | TBC | TO |
| 4 | C | TBT | TO | 13 | T | TBT | T |
| 5 | C | TB | C | 14 | T | TB | T |
| 6 | C | TBC | C | 15 | T | TBC | TO |
| 7 | SU | TBT | TO | 16 | RT | TBT | RT |
| 8 | SU | TB | SU | 17 | RT | TB | RT |
| 9 | SU | TBC | SU | 18 | RT | TBC | TO |

## 5. Simulation Results and Discussion

### 5.1. Simulation

Simulation 1: The simulation scenario assumes the lead car follows the Highway Fuel Economy Test Cycle (HWFET). The ego car equipped with ACC based on MPC travels with an initial velocity of 10 m/s. The relative distance between the two cars at the beginning is 100 m. In the simulation, the sampling time of the controller is 0.1 s. The headway time is constant and chosen as 1.4 s. The prediction horizon $p = 30$. MATLAB, Simulink, and Model Predictive Control toolbox [32,41] are used in the simulation. The speed of the ego car obtained from this simulation is shown in Figure 15 and is used as input to the EMS of the ego car.

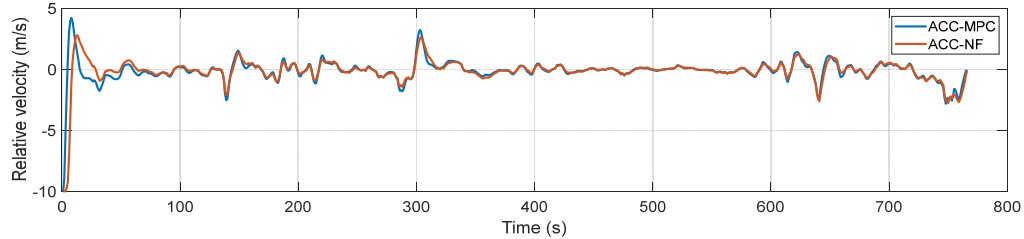

**Figure 15.** Relative velocity of between two cars.

Simulation 2: The simulation conditions for simulation 2 remain the same as simulation 1, but this time the ego car is equipped with ACC based on NF.

### 5.2. Discussion

Both ACC based on SMPC and neuro-fuzzy can guarantee safety for the ego car. The simulation results in Figure 16 show that safety is satisfied in two cases; spacing errors are always bigger than or equal to zero, so the collision is avoided.

The ego car behaviour is satisfied as to the tracking ability of velocity working well in both cases, and the relative distance is finally adjusted to the safe values. The ego car in both cases decelerates at the beginning, to avoid a collision with the lead car, and then regulates the velocity to adapt to the lead car's speed variation, while maintaining a safe distance computed by a constant time headway policy. The relative speed between these two vehicles is presented in Figure 15.

In terms of engine efficiency and the $SoC$ of the battery, by applying the EMS hinged on FLC, the motor torque and the engine torque of the ego car are produced, as shown in Figures 17 and 18, respectively. To date, there is no available commercial automotive computer simulator to simulate the algorithm; however, authors are working on the development of experimental analysis based on model/prototype and dimensional similitude

to show the concept as well as compare with a previous published model in [42]. It is expected that the method, results, and comparison are to be published in a follow up paper. Based on the engine efficiency map, the engine is operating in a region of sub-optimal fuel efficiency in the real time conditions. For this case, the average fuel efficiency is 28.70%, the torque is regulated mainly within a range of 10–84 (Nm), and the *SoC* of the battery during the whole trip is 0.7192, leading to the total fuel consumption of the vehicle being 6.74 L/100 km for the duration of a 16.5 km drive [42]. In simulation 1, with the amount of torque taken from the electric motor, the engine torque of the ego car is regulated within an interval of 32–80 (Nm). Consequently, the average energy efficiency of the engine is 31.52%, and the *SoC* at the end of the trip is 0.7236. Therefore, in terms of the total equivalent fuel consumption, the vehicle consumed 6.14 L/100 km for a distance of 16.5 km. Meanwhile, in simulation 2, the engine of the ego car falls in the same range of 32–80 (Nm). However, mainly during the trip, the torque of the engine is closer to the optimal torque area (as shown in Figure 18), making the total equivalent fuel consumption of the car being 5.98 L/100 km with an average fuel efficiency of 31.58% and the *SoC* at the end of the trip of 0.7286. Figures 19 and 20 represent the engine efficiency and *SoC* of the battery, respectively.

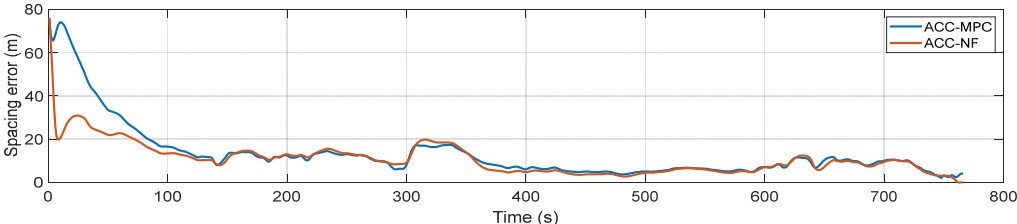

**Figure 16.** Spacing error between two vehicles.

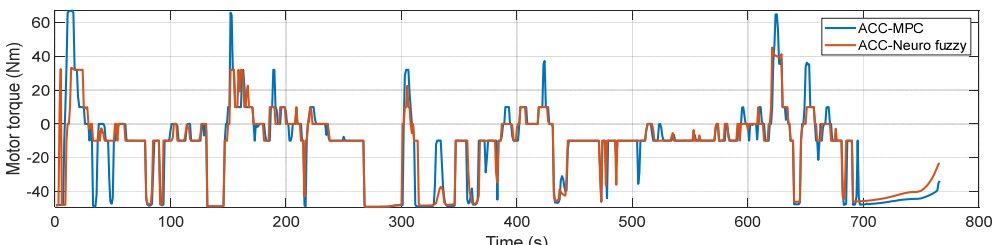

**Figure 17.** Motor torque of ego car.

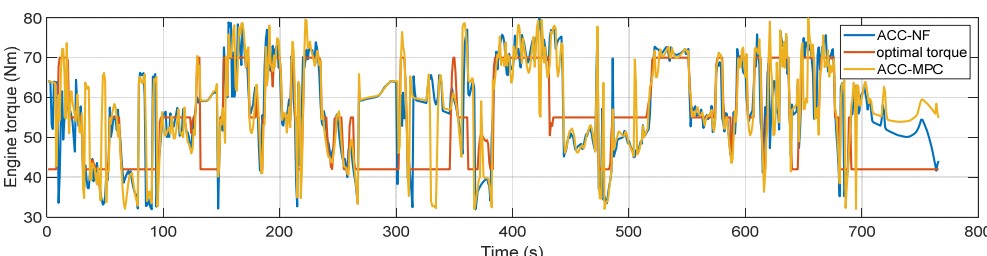

**Figure 18.** Engine torque of ego car.

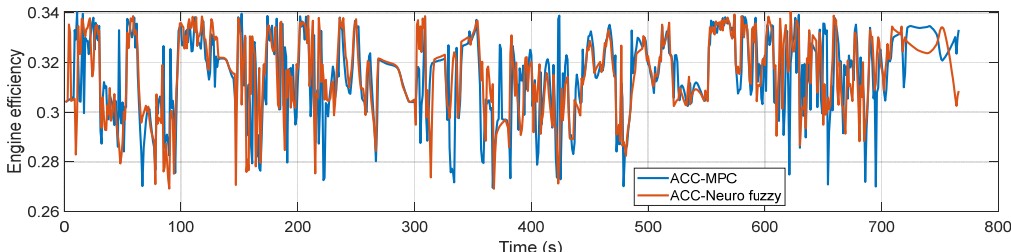

**Figure 19.** Engine efficiency of ego car.

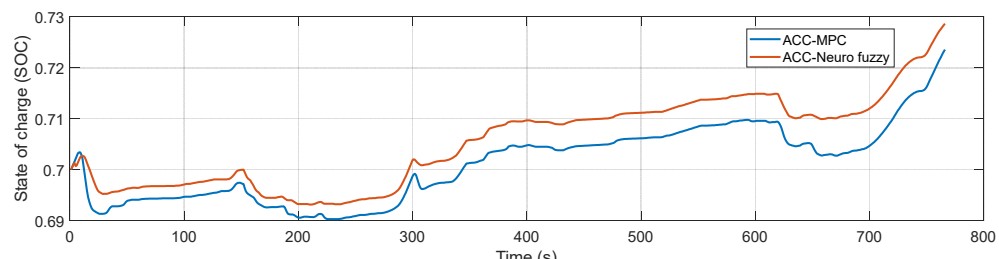

**Figure 20.** State of charge of battery of ego car.

## 6. Conclusions

In this paper, an intelligent driver assistance system for a HEAV is proposed to do two tasks: keeping the vehicle at a safe distance from the lead vehicle and reducing energy consumption. The system combined an ACC and EMS, in which the ACC is designed by combining a switched MPC technique with an NF approach. The performances of the proposed ACC systems are simulated and evaluated by considering the primary factors that affect vehicle performance, including vehicle specification, environment conditions, and driver behaviour. This paper shows that the driver assistance system can be improved by using the ACC-NF controller compared with using the ACC-SMPC controller. The system based on SMPC can generate excellent results. Still, the MPC needs to solve the optimization problem considering future prediction in every sampling, so it causes the burden calculation load. The NF method provides a practical alternative to conventional analytical control approaches in solving nonlinear autonomous control problems to obtain a robust, less computationally expensive, and more human-like speed control simultaneously. The simulation results demonstrate that the safe distance between two successive vehicles on the same are maintained during the trip. However, with the developed driver assistance system based on the ACC-NF method, the efficiency of the engine increases, and the *SoC* of the battery at the end of the trip is higher, leading to the equivalent fuel consumption of the vehicle improved by 2.61% compared with the system-based ACC-SMPC method and 8.9% compared with the system without any intelligent controller.

**Author Contributions:** Conceptualization, D.P.V. and H.K.; Data curation, Z.A.-S. and D.B.P.; Formal analysis, D.P.V.; Investigation, Z.A.-S. and D.P.V.; Methodology, D.P.V., A.M.A., S.S.S., D.B.P. and H.K.; Writing—original draft, Z.A.-S., D.P.V. and R.J.; Writing—review & editing, A.M.A., M.F. and H.K. All authors have read and agreed to the published version of the manuscript.

**Funding:** This research received no external funding.

**Acknowledgments:** The authors would like to thank Mahdi Jalili, for his helpful comments during this work.

**Conflicts of Interest:** The authors declare no conflict of interest.

## Nomenclature

| | |
|---|---|
| $\tau_e$ | Engine torque (N·m) |
| $\omega_e$ | Engine speed (rpm, rad/s) |
| $\tau_m$ | Motor torque (N·m) |
| $\omega_m$ | Motor speed (rpm·rad/s) |
| $\tau_{vehicle}$ | Torque of vehicle (N·m) |
| $\theta$ | Road inclination |
| $\rho$ | Air density, (kg/m$^3$) |
| $\eta_m$ | Mechanical efficiency |
| $\eta_e$ | Engine efficiency |
| $\dot{m}_{fuel}$ | Mass flow rate fuel consumption, kg/s |
| $\dot{m}_{eqv}$ | Equivalent fuel mass flow rate, kg/s |
| $\dot{Q}$ | Power |
| $q_c$ | Combustion energy (kJ/kg) |
| $C_{drag}$ | Drag coefficient |
| $C_{rolling}$ | Road friction coefficient |
| $C_T$ | Constant torque |
| $A(\phi)$ | Front surface area, m$^2$ |
| $AFI(\lambda)$ | Function of air to fuel ratio |
| $A/F$ | Air to fuel ratio |
| $P_{ma}$ | Manifold pressure |
| $P_b$ | Battery power (kW) |
| T | Intake temperature, °C |
| $T_{room}$ | Temperature of the air in the cabin, °C |
| $v_t$ | Speed of the vehicle at $t$ (m/s) |
| $v_w$ | Absolute wind speed (m/s) |
| $V_m$ | Manifold volume, m$^3$ |
| $V_{disp}$ | Volumetric displacement of the engine, m$^3$ |
| $SoC$ | State of Charge |
| MAX | Maximum flow through the throttle |
| HEV | Hybrid Electric Vehicle |
| MPC | Model Predictive Control |
| ACC | Adaptive Cruise Control |
| SMPC | Switching MPC |
| AMPC | Adaptive MPC |
| NF | Neuro Fuzzy |
| ANFIS | Adaptive-Network-based Fuzzy Inference System |

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
