# Peer review of "Intelligent Driver Assistance and Energy Management Systems of Hybrid Electric Autonomous Vehicles"

_sustainability, doi:10.3390/su14159378_

Round 1

Reviewer 1 Report

I gone through the paper, It is nicely written ( as per me).

However, I found the authors claim of improving the engine driving efficiency by 2.6 % which is based on simulation of Fuzzy based algorithm, which is not convincing to me until that claim is substantiated  by real time results or apt mathematical analysis/ proof. 

Author Response

The authors greatly appreciate the time and efforts that the anonymous reviewers have devoted to reviewing our manuscript; “Intelligent Driver Assistance and Energy Management Systems of Hybrid Electric Autonomous Vehicles”, Manuscript ID: sustainability-1758487. The manuscript has been revised based on the comments of the reviewers and the requested changes have been implemented in the manuscript and highlighted in red. The authors’ response to each comment has been outlined below.

Reviewer 2 Report

"In this study, the authors have studied the "Intelligent Driver Assistance and Energy Management Systems of Hybrid Electric Autonomous Vehicles."

The relevance and novelty of the manuscript are shown clearly.  The paper's subject matter is relevant to the scope of the journal. The introduction part highlights the importance of the topic. It is proposed to give a deeper insight into the PHEV opportunities as for example it is described in https://doi.org/10.1051/matecconf/201823500037 .

The abstract highlights the importance of fuel consumption reduction, emission decrease, and the importance of a sustainable approach, but I miss these aspects from the introduction part. These three elements should be more present here. The mentioned source contains a good description of fuel economy and has several references from the area.  The sustainability aspect is highlighted in several sources, for example in https://doi.org/10.55343/cogsust.7, challenges of autonomous vehicles (https://iopscience.iop.org/article/10.1088/1757-899X/252/1/012096/meta)

What is the reason behind, that the authors have chosen ANFIS system? Were there any other options?

Figure 6 and Figure 7 should be replaced.

Are there any experiences what is the correlation of the simulation results with a real-world validation? Could the same result be expected?

Author Response

(The authors gave the same response as above.)
